# Cancer-Associated Fibroblasts: Accomplices in the Tumor Immune Evasion

**DOI:** 10.3390/cancers12102969

**Published:** 2020-10-14

**Authors:** Marc Hilmi, Rémy Nicolle, Corinne Bousquet, Cindy Neuzillet

**Affiliations:** 1Department of Medical Oncology, Curie Institute, University of Versailles Saint-Quentin, 92210 Saint-Cloud, France; cindy.neuzillet@gmail.com; 2GERCOR, 151 rue du Faubourg Saint-Antoine, 75011 Paris, France; 3Programme Cartes d’Identité des Tumeurs (CIT), Ligue Nationale Contre Le Cancer, 75013 Paris, France; remy.nicolle@ligue-cancer.net; 4Cancer Research Center of Toulouse (CRCT), INSERM UMR 1037, University Toulouse III Paul Sabatier, ERL5294 CNRS, 31000 Toulouse, France; corinne.bousquet@inserm.fr; 5Institut Curie, Cell Migration and Invasion, UMR144, PSL Research University, 26, rue d’Ulm, F-75005 Paris, France

**Keywords:** cancer-associated fibroblasts, immunology, tumor microenvironment, cell communication

## Abstract

**Simple Summary:**

A growing number of studies suggest that cancer-associated fibroblasts (CAFs) modulate both myeloid and lymphoid cells through secretion of molecules (i.e., chemical function) and production of the extracellular matrix (ECM), i.e., physical function. Even though targeting functions CAFs is a relevant strategy, published clinical trials solely aimed at targeting the stroma showed disappointing results, despite being based on solid preclinical evidence. Our review dissects the interactions between CAFs and immune cells and explains how a deeper understanding of CAF subpopulations is the cornerstone to propose relevant therapies that will ultimately improve survival of patients with cancer.

**Abstract:**

Cancer-associated fibroblasts (CAFs) are prominent cells within the tumor microenvironment, by communicating with other cells within the tumor and by secreting the extracellular matrix components. The discovery of the immunogenic role of CAFs has made their study particularly attractive due to the potential applications in the field of cancer immunotherapy. Indeed, CAFs are highly involved in tumor immune evasion by physically impeding the immune system and interacting with both myeloid and lymphoid cells. However, CAFs do not represent a single cell entity but are divided into several subtypes with different functions that may be antagonistic. Considering that CAFs are orchestrators of the tumor microenvironment and modulate immune cells, targeting their functions may be a promising strategy. In this review, we provide an overview of (i) the mechanisms involved in immune regulation by CAFs and (ii) the therapeutic applications of CAFs modulation to improve the antitumor immune response and the efficacy of immunotherapy.

## 1. Introduction

Control and elimination of tumor cells by the immune system (also known as immunosurveillance) is a defense system that plays a major role in the prevention of cancers [1]. However, tumor growth in immunocompetent patients indicates that antitumor immunity can be dodged and no longer fulfill its full role. Indeed, modulating immunity toward tumor tolerance and evading the immune system are hallmarks of cancer [2,3].

The tumor microenvironment (TME) is actively involved in immune evasion leading to cancer progression and metastasis [4]. This compartment comprises several cell types, in addition to tumor cells, including pericytes, endothelial cells (venous, arterial and lymphatic), immune cells, nerve cells, adipocytes and cancer-associated fibroblasts (CAFs). CAFs are the most abundant cell type within the TME and play prominent roles by communicating with other cells and by secreting the extracellular matrix (ECM) components [5]. Similar to other cell types, recent studies have revealed that CAFs count multiple subpopulations with different functions [6]. This heterogeneity can be partially explained by the diverse origins of CAFs, ranging from local precursors including fibroblasts [5], pericytes [7], adipocytes [8], epithelial (epithelial-to-mesenchymal transition) [9] and endothelial cells (endothelial-to-mesenchymal transition) [10], to distant precursors such as bone marrow-derived mesenchymal cells [11] and circulating fibrocytes [12]. In noncancerous conditions, resident tissue fibroblasts are quiescent cells acting as sentinels and maintaining tissue homeostasis.

Upon tissue injury, such as in wound healing or in cancer, they differentiate into myofibroblasts (i.e., activated fibroblasts) to repair damaged tissue by secreting the ECM and by interacting with immune cells [13]. These functions are rewired by tumor cells, making CAFs their accomplices [5]. The discovery of the immunogenic role of CAFs has made their study particularly attractive due to potential applications in the field of cancer immunotherapy. A growing number of studies suggest that CAFs modulate both myeloid and lymphoid cells through secretion of molecules (i.e., chemical function) and production of ECM (i.e., physical function). In this review, we provide an overview of (i) the mechanisms involved in the immune regulation by CAFs, and (ii) the therapeutic applications of CAFs modulation to improve the antitumor immune response and the efficacy of immunotherapy.

## 2. CAFs Constitute A Chemical Immune Barrier

Once activated, CAFs produce several molecules to interact with immune cells, such as growth factors and cytokines [5]. Through these secretions, CAFs affect recruitment and characteristics of both myeloid (Table 1) and lymphoid cells (Table 2) to build a chemical immune barrier and, therefore, create an immunosuppressive TME favorable for cancer progression. CAFs activation is then perpetuated by immune cells through positive feedback loops. Among the cytokines that play a central role in the activation of fibroblasts, transforming growth factor β (TGFβ) is one of the most important [14]. The positive-feedback loop between CAFs and immune cells through TGFβ is well established; both CAFs and immune cells have the ability to secrete and respond to TGFβ [15].

### 2.1. Myeloid Cells

#### 2.1.1. Tumor-Associated Macrophages (TAMs)

TAMs comprise different subpopulations with distinct functionalities and derive from circulating monocytes or resident tissue macrophages. CAFs enrich the TME with TAMs by promoting monocytes precursors recruitment. In 2012, Ren, et al. showed that mesenchymal stromal cells from lymphoma, one of the precursors for CAFs, led to the intratumoral accumulation of CD11b+Ly6C+circulating monocytes by secreting chemokine-ligand 2 (CCL2) [17]. This chemokine binds to C-C chemokine receptor type 2 (CCR2) on monocytes and macrophages and attracts them to the tumor. Other preclinical studies suggested involvement of the CCR2-chemokine axis to TAM recruitment in breast cancer [19,20]. Once TAMs infiltrate tumor tissue, CAFs are able to differentiate them toward an M2-like phenotype. This phenotype is characterized by the promotion of Th2-type immune responses, angiogenesis and the inhibition of cytotoxic T lymphocytes (CTL) that lead to immune suppression and support cancer progression [48]. M2 differentiation is induced by CAF-secreted molecules such as Chitinase 3 Like 1 (Chi3L1), C-X-C motif chemokine 12 (CXCL12) and interleukin 6 (IL-6) in breast [16], prostate [21] and pancreatic cancers [22], respectively. Conversely, M1 differentiation is characterized by promotion of Th1-type immune responses and CTL activation. An increased M1/M2 ratio in cancer tissue is associated with a prolonged overall survival (OS) in cervical, ovarian and gastric cancers, and in lymphoma and myeloma [49].

#### 2.1.2. Myeloid-Derived Suppressor Cells (MDSCs)

MDSCs are immature myeloid cells that are only present in pathological conditions associated with chronic inflammation. In the case of cancer, they exert an immunosuppressive function by inhibiting T cell functions and promoting carcinogenesis [50]. Like TAMs, they are recruited by CAFs within the TME through the CCR2-chemokine axis as illustrated by Yang et al. [26] in the example of intrahepatic cholangiocarcinoma. Fibroblast activation protein (FAP) expression by CAFs activates fibroblastic STAT3 signaling leading to CCL2 secretion. Similar results were observed in colorectal and pancreatic cancers [24,29]. Other studies described additional mechanisms of the recruitment of MDSCs by CAFs involving the CXCL1 chemokine, followed by differentiation by CXCL12, IL-6, vascular endothelial growth factor (VEGF) and macrophage colony-stimulating factor (M-CSF) [28,29]. A single-cell RNA sequencing study conducted in prostate cancer demonstrated that different clusters of CAFs secreted various levels of proinflammatory cytokines such as CXCL12 and CCL2, highlighting that several CAF subpopulations may interact with MDSCs [51].

#### 2.1.3. Dendritic Cells (DCs)

The CTL antitumoral response requires the recognition of tumor epitopes presented by antigen presenting cells (APCs). APCs ingest tumor antigens and process them for presentation to T cells. Further, the activation of naive CD8+ T cells, leading to their proliferation and differentiation, also requires a costimulation signal in addition to antigen recognition. Once activated, tumor antigen-specific T cells are able to recognize and destroy pathologic cells presenting the same epitope. Tumor cells express major histocompatibility complex (MHC) class I molecules but often do not express costimulatory molecules. DCs can recognize antigens from a cell (e.g., tumoral) and activate another cell (e.g., CTL); this is called cross-priming [52]. DC modulation is a therapeutic avenue actively explored in immune-oncology [53]. CAFs can hijack DCs by recruiting them and deflect their function [23]. Indeed, De Monte, et al. [32] showed that pancreatic CAFs secreted thymic stromal lymphopoietin (TSLP) that conditioned myeloid DCs toward promotion of the protumorigenic Th2 response. CAFs modulate DCs to create an immunosuppressive TME through other mechanisms such as induction of regulatory hepatic DCs or inhibition of lung DC differentiation by IL-6 [30] and kynurenine secretion [31], respectively.

#### 2.1.4. Tumor-Associated Neutrophils (TANs)

TANs are associated with poor prognosis in several cancer types, including melanoma, renal, pancreatic, colorectal and gastric cancers [54]. Like TAMs, TANs can be roughly separated into two polarized populations: antitumorigenic N1 TANs, which are cytotoxic against tumoral cells, and protumorigenic N2 TANs, inhibiting CTL functions [55]. The interactions between CAFs and N2 TANs have not been explored yet, but studies in liver [33] and breast cancers [34] showed that CAFs recruit TANs using the same molecules as those involved in M2-reprogramming (i.e., CCL2, CXCL12 and IL-6).

#### 2.1.5. Mast Cells

Mast cells are tissue-resident quiescent cells acting as sentinels, releasing cytokines and chemokines when activated. They play an immunosuppressive role in the TME by secreting free adenosine and IL-13 that, respectively, inhibit CTL and activate M2 polarization, as well as promoting MDSCs and regulatory T cells (Tregs). Ellem, et al. [35] described that estrogen-activated prostatic CAFs secrete CXCL12 to recruit mast cells in a CXCR4-dependent manner.

### 2.2. Lymphoid Cells

#### 2.2.1. Cytotoxic T Lymphocytes (CTLs)

The main immune mechanism of tumor eradication is driven by CTLs specific for tumor antigens. Indeed, a strong intra-tumoral infiltration by CD3+/CD8+ CTLs, and CD45RO+ memory T lymphocytes is correlated with longer OS in most cancers [56]. Several studies have shown that CAFs are able to inhibit CTLs, either directly or indirectly. For example, Lakins et al. [41] showed that CAFs could kill and exclude CTLs from the TME by expressing programmed death ligand 2 (PD-L2) and Fas ligand in immunogenic tumors such as lung and melanoma. CTL exclusion is also promoted by the secretion of CXCL12 [37], TGFβ [38,39] and βig-h3 stromal protein [40] by CAFs. Moreover, CAFs can inhibit CTL cytotoxic activity and proliferation by producing immunosuppressive molecules such as nitric oxide [42] and adenosine [43] or by recruiting immunosuppressing cells such as MDSCs [24,25,29] or regulatory DCs [30].

#### 2.2.2. Helper T Lymphocytes

As mentioned above, an enrichment in cytotoxic and memory T-lymphocytes is associated with a favorable prognosis. This infiltration is associated with a Th1-type immune orientation of CD4+ T lymphocytes involving T-bet, STAT-1 and IRF1 transcription factors leading to the local production of IL-12 and interferon γ. In order to counteract this antitumor immune phenotype, CAFs promote a Th2 orientation, mutually exclusive of Th1, leading to the secretion of the immunosuppressive IL-10 cytokine. This Th2 phenotype can be induced either by CAF-secreting Chi3L1, or by orientating DCs to secrete Th2 chemokines, such as IL-13, in breast [16] and pancreatic cancers [32], respectively. Another CAF subset was recently discovered with the ability to present antigens to CD4+ T cells by expression of the MHC class II [57]. However, the absence of costimulatory molecules deactivates CD4+ T cells and decreases the CD8+ T cells/Tregs ratio [57] thus inhibiting the antitumor response.

#### 2.2.3. Regulatory T Cells (Tregs)

Tregs are CD4+ T cells whose function is to inhibit or attenuate immune responses. Most of Tregs strongly express CD25 (alpha chain of the IL-25 receptor) and the transcription factor FoxP3. The prognostic value associated with a high Treg density within the tumor is controversial [58]. A negative prognostic value has been described in pancreatic [59], ovarian [60], liver [61] and breast cancers [62], whereas a positive prognostic value has been observed in lymphomas [63], head and neck [64] and colorectal cancers [65]. There are many reasons for these conflicting observations, such as an insufficiently in-depth analysis of this population requiring consideration of their functionality and subpopulations; hence, FoxP3 expression is not limited to Tregs [66] and Tregs may lose FoxP3 expression [67]. Besides, the favorable prognosis associated with Treg infiltrates may be due to the concomitant presence of effector T cells. As CTLs, Tregs can be recruited and expanded by CAFs either directly by CXCL12, OX40L, PD-L2, and JAM2 expression [44,45], or indirectly through MDSCs and regulatory DCs recruitment [25,30]. A recent study by Kieffer et al. [68] used a single-cell RNA sequencing approach to analyze intratumoral heterogeneity and showed that FAP+ CAFs with an ECM signature drive immunosuppression in several cancer types by recruiting Tregs.

#### 2.2.4. Natural Killer Cells (NK Cells)

NK cells, or type I innate lymphoid cells, are lymphocytes belonging to innate immunity and do not express antigen receptors clonally synthesized by B and T lymphocytes. NK cells exert an antitumor activity by secreting pro-Th1 cytokines or by directly killing tumoral cells through the release of cytotoxic granules containing perforins and granzymes [69]. Previous studies showed that a high cytotoxic activity of NK cells correlated with a prolonged OS in head and neck cancers [70] while a decrease in the cytotoxic activity was associated with shorter OS and the occurrence of metastases in gastric [71], liver [72] and colorectal cancers [73]. The study of tumoral NK cells revealed functional and phenotypic alterations, such as reduced cytotoxicity, decreased production of antitumoral IFNγ and tumor necrosis factor α (TNFα) cytokines, and low expression of activating receptors [74]. CAFs are responsible for these functional defects by secreting prostaglandin E2 (PGE2) and indoleamine 2,3-dioxygenase (IDO) in melanoma [46] and hepatocellular carcinoma [47].

## 3. CAFs Constitute a Physical Immune Barrier

CAFs secrete excess deposits of collagenous and noncollagenous ECM along with degradation enzymes that promote metastasis and cancer progression [75]. Besides the direct effect on cancer cells [76], ECM-remodeling also contributes to modulation of immune cells by creating a physical immune barrier. Indeed, mouse models of pancreatic adenocarcinoma have shown that T cells lose their ability to infiltrate the tumor area when the ECM density is high [77]. Enriched type I collagen ECM activates the expression of leucocyte inhibitory receptors such as LAIR-1 and decreases T cell infiltration [78]. In the ECM of melanoma, the loss of HAPLN1 secretion (hyaluronic protein) by aged fibroblasts induces protumorigenic effects by inhibiting T cell motility while increasing that of polymorphonuclear immune cells, which, in turn, recruit Tregs [79]. Furthermore, ECM stiffness may also impact the innate immunity. Preclinical studies suggest that macrophage polarization and migration is affected by the ECM physical properties [80]. ECM alterations in cancer, such as increased stiffness, type I collagen and hyaluronan, promote infiltration of motile M2 macrophages [81,82,83]. In addition to ECM stiffness and composition, Salmon et al. showed that the fibers’ orientation controls the migration of T cells so that aligned fibers around vascular and tumoral regions exclude T cells from lung tumor islets [84]. Besides, the recruitment of immunosuppressive innate immune cells by the ECM can also result from indirect interactions. For example, the high-density of the ECM in breast and pancreatic cancers activates tumor cells and CAFs that secrete the monocyte cytokines CCL2 and colony-stimulating factor 1 (CSF-1) [85,86].

Finally, CAF-induced fibrosis generates a high interstitial pressure leading to a poorly vascularized, hypoxic TME, which constrains access to cell nutrients and, thus, cellular phenotype and metabolism. Moreover, a positive-feedback loop exists since fibrosis is enhanced in hypoxic regions [87]. Indeed, hypoxia activates CAFs to secrete type I collagen (i.e., immune physical barrier) and immunosuppressive cytokines such as IL-12 and CXCL12 (i.e., immune chemical barrier) [88]. Besides, activation of hypoxia-inducible factor-1α (HIF-1α) by hypoxia plays a central role in creating an immunosuppressive TME. First, HIF-1α upregulates the expression of PD-L1 by MDSCs, macrophages, DCs, and tumor cells, thereby promoting T cell inactivation through the PD-1/PD-L1 axis [89]. Secondly, HIF-1α impacts T cell activity by increasing NF-κB activation, which decreases the transduction of the cell receptor (TCR) signal [90]. Thirdly, hypoxia triggers the CD39 and CD73 ectonucleotidases, generating extracellular formation of immunosuppressive adenosine [91]. Furthermore, there is a positive-feedback loop between recruited immunosuppressive macrophages and CAFs, with macrophages activating CAFs by granulin secretion [86,92]. Overall, the CAFs-induced ECM constitutes a physical immune barrier by impeding both adaptive and innate immunity directly through mechanical forces or indirectly by promoting an immunosuppressive hypoxic microenvironment.

## 4. Good Cop CAFs: Looking for A Needle in A Haystack

The literature predominantly supports the tumor-promoting role of CAFs, but there is also evidence that some CAF subsets restrain tumor activity. This is particularly important because therapeutic ablation of antitumoral CAFs can be deleterious. Indeed, nonspecific depletion of α-SMA+ myofibroblasts and SHH-dependent CAFs in pancreatic cancer generated more aggressive tumors (poorly differentiated, more vascularized and infiltrated by Tregs) worsening patient survival [93,94,95]. Current CAFs markers such as α-SMA, vimentin, FAP, platelet-derived growth factor receptor α (PDGFRα) and FSP1 are nonexclusive for fibroblasts and expressed across multiple CAFs subtypes [96]. Several studies in many cancer types deciphered CAFs subsets according to their phenotype and function in order to target appropriate subpopulations and ultimately sensitize tumors to systemic therapies (i.e., immunotherapies and chemotherapies) [44,97].

Identifying tumor-restraining CAFs is not an easy task. First, studies can be conflicting regarding the same marker. As an example, podoplanin-expressing CAFs were shown to be associated with good prognoses in breast [98] and colon cancers [99] but the opposite finding was reported in breast [100,101,102] and lung cancers [103,104]. Secondly, the impact of CAFs on tumor cells varies according to the cancer type. Haro et al. highlighted that a stromal signature representing CAFs and ECM components was predictive of good survival in B lymphomas and decreased survival in carcinomas [105]. The authors supported the hypothesis that the immune suppressive functions of CAFs, notably on B cells through the TGF-β pathway, disrupted tumorigenesis in B lymphomas while promoting cancer progression in carcinomas. Similarly, hedgehog-signaling CAFs are associated with reduced colon cancer progression [106] but act as enhancers of pancreatic cancer progression [107,108].

To date, the exact phenotypes of good tumor-restraining CAFs remains unknown. Some studies suggested that normal quiescent fibroblasts resistant to activation may suppress tumorigenesis and metastasis [109,110,111]. Other studies showed the presence of active antitumor CAFs that can directly act on tumor cells by inhibiting their growth and phenotype transformation [109,112,113]. Several single cell RNA sequencing studies have uncovered CAF heterogeneity in pancreatic [97,114,115,116], head and neck [117], breast [118,119], lung [120], colorectal [121] cancers and melanoma [122]. However, CAF functionality hypothesized from the transcriptomic profiles requires further experimental validations. Indeed, there is a need to functionally assess and validate CAF functions in vitro and in vivo, but this is challenging since isolating and growing pure subpopulations of CAFs without disturbing their functionality is challenging [123].

## 5. Therapeutic Implications

It is now acknowledged that increasing tumor immunogenicity is relevant to induce cancer regression and prolonged survival in cancer patients [124]. Considering that CAFs are TME orchestrators and modulate immune cells, targeting their functions may be a promising strategy. As depleting all CAFs is harmful, identifying and selectively targeting the deleterious pathways and subpopulations remains the main avenue of research. However, this is a hard task since cells within the TME interact with each other in a complex network. Here, we will develop current strategies aimed at modulating CAF functions to improve tumor immunogenicity. Since fibrosis is an immune physical barrier, desmoplasia depletion may sensitize tumor cells to immune attack and several strategies are underway to achieve this goal (Figure 1).

The first strategy is to reduce ECM production. The clinical use of all-trans retinoic acid (ATRA) is well established in acute promyelocytic leukemia for its differentiating effect. ATRA has also been associated with ECM suppression and pancreatic stellate cells (pancreatic resident fibroblasts) inhibition (i.e., quiescence induction) [125]. ATRA is currently explored in early phase trials in melanoma, pancreatic, breast and prostate cancers in combination with other drugs including immune therapies (NCT04241276, NCT04113863, NCT03572387, NCT02403778). Similarly, vitamin D receptors are major transcriptional factors to get pancreatic stellate cells back to a quiescent state and reduce ECM remodeling [126]. Paricalcitol, a vitamin D receptor ligand, was evaluated in metastatic pancreatic cancer in early phase trials in association with chemotherapy (NCT03520790, NCT03415854) and yielded a 83% objective response rate in combination with chemotherapy and nivolumab in a phase II pilot trial [127], leading an expansion cohort (NCT02754726). Furthermore, normalization of the deleterious high interstitial pressure and resulting hypoxia could be achieved with the systemic administration of PEG-fused hyaluronidase (PEHPH20) [128]. The clinical development of PEHPH20 has mainly been carried out in pancreatic cancer. An efficacy signal was detected in combination with gemcitabine plus nab-paclitaxel in hyaluronan-high tumors in a randomized phase II study including 279 patients with untreated metastatic PDAC [129]. However, PEGPH20 failed to improve OS in the HALO 301 phase III trial [130]. Moreover, a phase Ib/II trial showed a negative effect on OS and additional digestive toxicities when PEGPH20 was added to FOLFIRINOX (5-fluorouracil, leucovorin, irinotecan and oxaliplatin) in pancreatic cancer [131]. Some trials are ongoing in resectable and advanced pancreatic cancer in combination with immune checkpoints inhibitors (ICI) (NCT03634332, NCT03979066), following evidence that PEGPH20 can promote T cell infiltration into the TME [132]. Additional ECM targets, such as matrix metalloproteinases (MMPs) inhibitors, showed disappointing clinical results with severe musculoskeletal toxicities [133,134,135], despite encouraging preclinical data [136,137]. Besides, glutamine is involved in hyaluronan synthesis through the hexosamine biosynthesis pathway [138] and its antagonism increases TILs and sensitizes anti-PD1 therapy in preclinical models [139]. Glutamine antagonism was limited by gastrointestinal toxicities in previous clinical trials owing to overdosing and is now redeveloped in a low dose regimen to avoid significant toxicity [140]. Furthermore, the focal adhesion kinase (FAK) signaling pathway regulates fibrosis [141] but also has immunomodulatory functions through Treg recruitment [142]. Its inhibition by a tyrosine kinase inhibitor, defactinib, was evaluated in advanced solid tumors in combination with anti-PD1 therapy (NCT0254653, NCT03727880, NCT02758587) based on preclinical evidence. Moreover, IL-1 mediated signaling promotes CAF proliferation and fibrosis [143]. The IL-1 receptor blockade has shown interesting efficacy results (disease control rate 84%) in metastatic colorectal cancer in combination with 5-fluorouracil and bevacizumab [144] in a single-arm phase II study, and is now being explored in pancreatic cancer (NCT02021422). Additionally, other antifibrosis molecules approved for lung fibrosis are currently being evaluated in oncology following the demonstration of their immunomodulatory properties in animal models. Pirfenidone has been shown to reduce lung cancer growth in murine models by down-regulating the TGFβ pathway leading to an increase in NK and T cells infiltration [145] and giving a rationale for combination with atezolizumab in patients with lung cancer (NCT04467723). Nintedanib is a tyrosine kinase inhibitor that targets drivers of fibrosis and angiogenesis such as vascular endothelial growth factor receptors, fibroblast growth factors receptors, and platelet-derived growth factors receptors [146]. This drug inhibited activation of protumoral lung CAFs in preclinical models [147] and is currently being evaluated in patients with lung cancer in combination with nivolumab and ipilimumab (NCT03377023). Losartan is an approved angiotensin II receptor inhibitor to treat high blood pressure and also inhibits collagen I synthesis [148]. A single-arm phase II trial evaluating its association with chemotherapy showed interesting downstaging rates (R0 resection rate 61%) in 49 patients with pancreatic cancer [149]. A randomized phase II is ongoing in combination with chemotherapy, immunotherapy and radiotherapy in patients with localized pancreatic cancer (NCT03563248).

Finally, the hedgehog signaling pathway involving the Sonic Hedgehog (SHH) proteins promoted desmoplasia [107] and stimulated stellate cell differentiation and myofibroblast activation in pancreatic cancer. SHH inhibition showed promising results in mouse models but did not improve OS in clinical trials [150,151]. As hedgehog-signaling CAFs are heterogenous and encompass several subsets [6], one hypothesis for this failure is the absence of targeting of protumoral CAF subsets. This second strategy has been explored by targeting protumoral CAF markers. As an example, FAP is linked to angiogenesis and immunosuppression [152] and its inhibition by monoclonal antibodies or small molecules was clinically inefficient in pancreatic [153], lung [154] and colorectal cancer [155]. Based on data showing immune control of tumor growth and effectiveness of immune checkpoint inhibitors after FAP inhibition in a KPC mouse model [37], FAP inhibition is currently being explored in association with pembrolizumab (anti-PD1) (NCT04171219, NCT03910660, NCT04007744). RO 6874281 is a recombinant fusion protein composed of a human monoclonal antibody directed against FAP linked to a variant form of IL-2. This drug stimulates a local immune response by the accumulation of CTLs and NK cells in FAP-expressing areas and is currently under evaluation in combination with anti-PD1 (NCT04171219, NCT03910660, NCT04007744), and with trastuzumab (anti-HER2) or cetuximab (anti-EGFR) in head and neck, and breast cancers (NCT02627274). Similarly, PDGFRα is one of the CAF markers associated with protumorigenic properties [109], which can be targeted by olaratumab. Olaratumab is approved for the treatment of soft tissue sarcoma [156] and is being investigated in pancreatic cancer (NCT03086369). Moreover, inhibition of NAD(P)H Oxidase-4 (NOX4) was able to revert protumoral CAF subsets such as myofibroblastic [157] and immune-suppressive CAFs [158] in preclinical models making its clinical development promising. A recent study found a new CAF lineage associated with poor response to anti-PD-L1 therapy across six cancer types by using single-cell transcriptomics [116]. This subpopulation expresses the leucine-rich repeat containing 15 (LRRC15) protein, is driven by TGFβ and could represent a new target in combination with immunotherapy.

The third therapeutic approach is to target the CAF-secreted molecules that promote the immunosuppressive TME, such as CCL2, CXCL12 and IDO. As mentioned above, the CCR2-CCL2 axis is involved in the recruitment of M2 TAM, and its blockade consequently reduces TAM infiltration [159]. However, no objective responses were observed with CCR2-CCL2 inhibitors when administrated as monotherapy in metastatic castration-resistant prostate cancer [160] or in combination chemotherapy in pancreatic cancer [161,162]. Despite these disappointing results, murine models showed that the CCR2-CCL2 axis blockade may potentiate ICI efficacy [163] leading to exploration of the combination with nivolumab (anti-PD1) in early phase trials in several cancer types (NCT03496662, NCT03767582, NCT03184870, NCT04123379). Furthermore, CXCL12-CXCR4 signaling is involved in stromal-immune crosstalk [164] giving a rationale for the clinical investigation of CXCR4 inhibitors in combination with anti-PD1 in metastatic pancreatic (NCT04177810, NCT02826486) and head and neck cancers (NCT04058145). In addition, since IDO is secreted by CAFs, and is involved in local immunosuppression by inducing NK defects, T-cell apoptosis and Treg activity, its inhibition is being actively explored in oncology. The first efficacy results showed no clinical benefit in melanoma [165] when an IDO inhibitor was added to anti-PD1, but objective response rates were up to 40–60% in head and neck, kidney and breast cancers [166]. Finally, the TGFβ secreted by tumor cells can turn CAFs into an inflammatory phenotype producing the protumoral and immunosuppressive IL-6 cytokine. Targeting the TGFβ pathway is being evaluated in ongoing clinical trials in several cancer types (NCT02423343, NCT02452008, NCT02581787, NCT03834662, NCT02937272, NCT02423343). As TGFβ is associated with poor response to ICI, its inhibition by galunisertib in combination with durvalumab (anti-PD-L1) was tested in metastatic pancreatic cancer and showed an acceptable safety profile [167]. Bintrafusp alfa is a bifunctional fusion protein composed of the TGF-β receptor combined to an antibody blocking PD-L1 with a manageable safety profile and encouraging activity in phase I trials in pretreated advanced solid tumors [168,169]. However, prediction of efficacy is difficult regarding both the antitumoral and protumoral effects of TGFβ signaling.

Overall, combining selective stromal modulation with ICI is being actively explored in early phase trials (Table 3) since the stroma-only modulation appeared disappointing. Many of the abovementioned drugs aimed at targeting the TME do not necessarily block CAF signals selectively but inhibit a specific target from multiple sources including cancer cells and recruited inflammatory cells. Although CAFs certainly affect the immune response, it is still unknown whether targeting both tumor-derived and CAF-derived signals would be more beneficial than an antitumor strategy targeted toward one specific cell type, and more work needs to be done to differentiate these two sources. Furthermore, it is still unknown if CAF immunomodulatory functions are organ-specific or pan-tumor since studies are usually conducted in one cancer type. However, single-cell studies showed that CAFs may share common characteristics independently of the primary cancer [68,116]. 

## 6. Conclusions

CAFs are highly involved in tumor immune evasion by physically and chemically impeding the immune system and interacting with both myeloid and lymphoid cells. However, CAFs do not represent a single cell entity but are divided into several subtypes with different functions that may be antagonistic. Targeting CAFs functions is a promising strategy currently being explored in early phase trials. However, published clinical trials solely aimed at targeting the stroma showed disappointing results, despite being based on solid preclinical evidence. This shows that a deeper understanding of CAF subpopulations and heterogeneity in the context of immune evasion is the cornerstone to propose relevant therapies that will ultimately improve survival of patients with cancers [123]. To achieve this goal, collaboration between clinicians and researchers is mandatory.

## Figures and Tables

**Figure 1 cancers-12-02969-f001:**
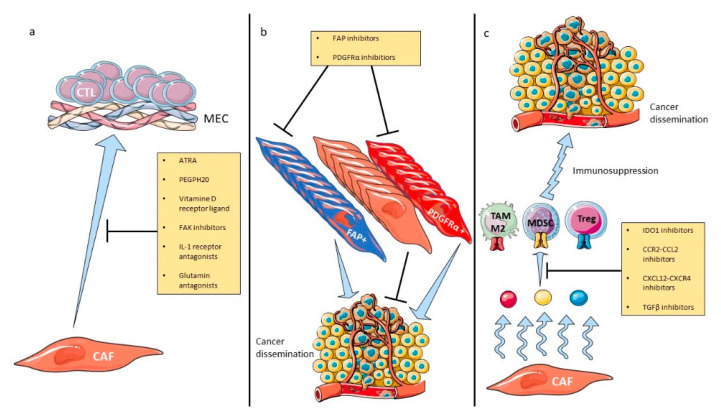
Therapeutic strategies modulating CAF activities. CAF-related drugs, with or without ICI combination, inhibit fibrosis (**a**), CAF-protumoral subsets (**b**) or CAF-secreted immunosuppressive molecules (**c**). Abbreviations: CAF: cancer-associated fibroblasts, CCL2: chemokine-ligand 2, CCR2: C-C chemokine receptor type 2, CXCL12: C-X-C motif chemokine 12, CXCR4: C-X-C motif chemokine receptor 4, ECM: extracellular matrix, FAK: focal adhesion kinase, FAP: fibroblast activation protein, ICI: immune checkpoint inhibitors, IDO1: indoleamine 2,3-dioxygenase 1, MDSC: myeloid-derived suppressor cells, PDGFRα: platelet-derived growth factor receptor α, TAM: tumor-associated macrophage, TGFβ: transforming growth factor β, Treg: regulatory lymphocyte.

**Table 1 cancers-12-02969-t001:** Summary of studies evaluating modulation of myeloid cells by CAFs ^1^.

Myeloid Cells	Effect	CAF-Secreted Molecules	Cancer Type	Study
TAM	Recruitment Reprogramming to an M2-like phenotype	Chi3L1	Breast	Cohen et al. [16]
	Recruitment	CCL2	Lymphoma	Ren et al. [17]
	Recruitment	NA	Pancreas	Gunderson et al. [18]
	Recruitment	CCL2	Breast	Jia et al. [19]
	Recruitment	CCL2	Breast	Ksiazkiewicz et al. [20]
	Recruitment Reprogramming to an M2-like phenotype	CXCL12	Prostate	Comito et al. [21]
	Reprogramming to an M2-like phenotype	IL-6 and IL-10	Pancreas	Mathew et al. [22]
	Recruitment	NA	Breast	Liao et al. [23]
MDSC	Recruitment	CCL2	Colorectal	Chen et al. [24]
	Recruitment	IL-6	Skin (squamous)	Ruhland et al. [25]
	Recruitment	CCL2	Biliary tract	Yang et al. [26]
	Recruitment	CXCL1	Lung, colon, melanoma, breast, pancreas, thymoma	Kumar et al. [27]
	Recruitment	NA	Breast	Liao et al. [23]
	Recruitment Differentiation	CXCL12	Liver	Deng et al. [28]
	Differentiation	IL-6, VEGF, M-CSF, CXCL12 and CCL2	Pancreas	Mace et al. [29]
DC	Recruitment	NA	Breast	Liao et al. [23]
	Induction of regulatory DC	IL-6	Liver	Cheng et al. [30]
	Inhibition of differentiation	Kynurenine	Lung	Hsu et al. [31]
	Induction of DC promoting Th2 polarization	TSLP	Pancreas	De Monte et al. [32]
TAN	Recruitment	CCL2	Lymphoma	Ren et al. [17]
	Recruitment, survival and activation	CXCL12 and IL-6	Liver	Cheng et al. [33]
	Recruitment	CXCL1, CXCL2 and CXCL5	Breast	Yu et al. [34]
Mast cells	Recruitment	CXCL12	Prostate	Ellem et al. [35]

^1^ CAF: cancer-associated-fibroblast. CCL: chemokine-ligand. CXCL: C-X-C motif chemokine. DC: dendritic cells. IL: interleukin. M-CSF: macrophage colony-stimulating factor. MDSC: myeloid-derived suppressor cells. NA: not available. TAM: tumor-associated macrophages. TAN: tumor-associated neutrophils. TSLP: thymic stromal lymphopoietin. VEGF: vascular endothelial growth factor.

**Table 2 cancers-12-02969-t002:** Summary of studies evaluating modulation of lymphoid cells by cancer-associated fibroblasts ^1^.

Lymphoid Cells	Effect	CAF-Induced Mechanism	Cancer Type	Study
CTL	Exclusion	Chi3L1 secretion	Breast	Cohen et al. [16]
	Inhibition of cytotoxicity	MDSC recruitment	Colorectal	Chen et al. [24]
	Inhibition of cytotoxicity and activation	MDSC recruitment	Skin (squamous)	Ruhland et al. [25]
	Exclusion	NA	Breast	Liao et al. [23]
	Inhibition of cytotoxicity	IDO-producing regulatory DC	Liver	Cheng et al. [30]
	Inhibition of proliferation	MDSC differentiation	Pancreas	Mace et al. [29,36]
	Exclusion	CXCL12 secretion	Pancreas	Feig et al. [37]
	Exclusion	TGFβ secretion	Urothelial	Mariathasan et al. [38]
	Exclusion	TGFβ secretion	Colon	Tauriello et al. [39]
	Exclusion	βig-h3 stromal protein	Pancreas	Goehrig et al. [40]
	Killing Exclusion	PD-L2 and FASL expression	Lung and melanoma	Lakins et al. [41]
	Inhibition of proliferation	NO secretion	Breast	Cremasco et al. [42]
	Inhibition of proliferation, activation and cytotoxicity	Production of adenosine	Cervical	De Lourdes Mora-García et al. [43]
Helper T lymphocytes	Promotion of Th2 phenotype	Chi3L1 secretion	Breast	Cohen et al. [16]
	Promotion of Th2 phenotype	NA	Breast	Liao et al. [23]
	Promotion of Th2 phenotype	DC secretion of Th2 chemokines	Pancreas	De Monte et al. [32]
Treg	Recruitment	MDSC recruitment	Skin (squamous)	Ruhland et al. [25]
	Recruitment	NA	Breast	Liao et al. [23]
	Expansion	IDO-producing regulatory DC	Liver	Cheng et al. [30]
	Recruitment Retention Differentiation	CXCL12 (recruitment) OX-40L, PD-L2, JAM2 (retention) B7H3, CD73, DPP4 (differentiation)	Breast	Costa et al. [44]
	Recruitment, survival and differentiation	CXCL12 secretion	Ovary	Givel et al. [45]
NK cells	Inhibition of cytotoxicity and cytokine production	PGE2 secretion	Melanoma	Balsamo et al. [46]
	Inhibition of cytotoxicity and cytokine production	PGE2 and IDO secretion	Liver	Li et al. [47]

^1^ Abbreviations. CAF: cancer-associated-fibroblast. CCL: chemokine-ligand. CXCL: C-X-C motif chemokine. CTL: cytotoxic T lymphocytes. DC: dendritic cells. JAM2: Junctional Adhesion Molecule 2. IDO: indoleamine 2,3-dioxygenase. MDSC: myeloid-derived suppressor cells. TGF: transforming growth factor β. PD-L2: Programmed death-ligand 2. PGE2: prostaglandin E2. VEGF: vascular endothelial growth.

**Table 3 cancers-12-02969-t003:** Ongoing clinical trials evaluating modulation of cancer-associated fibroblasts in combination with ICI in solid tumors ^1^.

Strategy	Target	Molecule	ICI	Phase	Solid Tumors	Population	ClinicalTrial.gov Reference
Inhibition of protumoral CAF subsets	FAP	CAR-T cell	Pembrolizumab	II	All	Advanced	NCT02546531
			Pembrolizumab	I/II	Prostate	Metastatic castration-resistant	NCT03910660
		Sonidegib	Pembrolizumab	I	All	Advanced	NCT02758587
Inhibition of CAF-secreted immunosuppressive molecules	CXCL12/CXCR4 axis	AMD3100	Cemiplimab	II	Pancreas	Metastatic	NCT04177810
			Pembrolizumab	II	Head and neck	Recurrent, metastatic	NCT04058145
		BL-8040	Pembrolizumab	II	Pancreas	Metastatic	NCT02826486
		Ulocuplumab	Nivolumab	I/II	All	Advanced	NCT02472977
	CCR2/CCL2 axis	BMS-813160	Nivolumab	I/II	Pancreas	Borderline, locally advanced	NCT03496662
			Nivolumab	I/II	Pancreas	Locally advanced	NCT03767582
			Nivolumab	I/II	All	Advanced	NCT03184870
			Nivolumab	II	Lung, liver	Resectable	NCT04123379
	IDO1	Epacadostat	Pembrolizumab	II	Bladder	Muscle-invasive	NCT03832673
			Pembrolizumab	I/II	All	Advanced	NCT02959437
			Pembrolizumab	II	Lung	Metastatic	NCT03322540
			Pembrolizumab	II	Lung	Metastatic	NCT03322566
			Pembrolizumab	I/II	All	Metastatic	NCT03085914
			Nivolumab	I/II	All	Advanced	NCT03347123
			Durvalumab	I/II	All	Advanced	NCT02318277
			Atezolizumab	I	Lung, bladder	Advanced	NCT02298153
			Pembrolizumab	II	Pancreas	Metastatic	NCT03006302
		BMS-986205	Nivolumab	II	Bladder	Non-muscle invasive	NCT03519256
			Nivolumab	I/II	Bladder	Muscle-invasive	NCT03661320
			Nivolumab, ipilimumab	I/II	All	Advanced	NCT02658890
			Nivolumab	II	Head and neck	Localized, Metastatic	NCT03854032
			Nivolumab	I/II	Liver	Advanced	NCT03695250
	TGFβ	Galunisertib	Nivolumab	I/II	All	Advanced	NCT02423343
Inhibition of fibrosis	FAK	Defatinib	Pembrolizumab	I	All	Advanced	NCT02546531
			Pembrolizumab	II	Pancreas	Resectable	NCT03727880
			Pembrolizumab	II	All	Advanced	NCT02758587
	CAF precursor	ATRA	Ipilimumab	II	Melanoma	Advanced	NCT02403778
		Paricalcitol	Nivolumab	II	Pancreas	Advanced	NCT02754726
	Hypoxia	PEGPH20	Pembrolizumab	II	Pancreas	Metastatic	NCT03634332
			Atezolizumab	II	Pancreas	Resectable	NCT03979066
	IL-1	Isunakinra	Unknown	I/II	All	Advanced	NCT04121442
		Canakinumab	Pembrolizumab	II	Lung	Resectable	NCT03968419
			Pembrolizumab	III	Lung	Advanced	NCT03631199
			Spartalizumab	I	Renal	Localized	NCT04028245
	TGFβ	Pirfenidone	Atezolizumab	I/II	Lung	Advanced	NCT04467723
	VEGFR, FGFR, PDGFR	Nintedanib	Nivolumab and Ipilumab	I/II	Lung	Advanced	NCT03377023
	Collagen	Losartan	Nivolumab	II	Pancreas	Localized	NCT03563248

^1^ Abbreviations. ATRA: all-trans retinoic acid. CCL2: chemokine-ligand 2. CCR2: C-C chemokine receptor type 2. CXCL12: C-X-C motif chemokine 12. CXCR4: C-X-C motif chemokine receptor 4. ECM: extracellular matrix. FAK: focal adhesion kinase. FAP: fibroblast activation protein. FGFR: fibroblast growth factor receptor. ICI: immune checkpoint inhibitors. IDO1: indoleamine 2,3-dioxygenase 1. IL-1: interleukin-1. PDGFR: platelet-derived growth factor receptor. TGFβ: transforming growth factor β. VEGF: vascular endothelial growth factor receptor.

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
