# Peer review of "Cancer-Associated Fibroblasts: Accomplices in the Tumor Immune Evasion"

_cancers, 2020, doi:10.3390/cancers12102969_

Round 1

Reviewer 1 Report

It is a precise and well designed description of the role of the CAF in the context of tumor immune evasion and the immune therapeutic strategies in progress. The only remark concerns he signification of the term NA in table 1 which is not defined.

Author Response

Paris, 4th October 2020

Dear Reviewer,

We would like to thank you for your interest in our work and your positive comments.

It is a precise and well designed description of the role of the CAF in the context of tumor immune evasion and the immune therapeutic strategies in progress. The only remark concerns he signification of the term NA in table 1 which is not defined.

We defined the term NA in Table 1.

We also made a detailed check for language errors throughout the manuscript.

Best regards,

Marc Hilmi

Reviewer 2 Report

This review article about immune modulation by CAF summarizes some of the interactions that lead to immune evasion by the TME. He manuscript is well presented and highlights the potential mechanisms (immune and chemical barrier) and immune/inflammatory cells involved in this process. In the final section the authors provide a list of current therapeutic approaches in clinical trials.

  • LN 37,38. The tumor microenvironment has more than four cell types, several groups also highlighted the role of nerves, adipocytes and lymphatics in the TME.
  • A small discussion about the effects of the immune/inflammatory cells on perpetuating CAF activation is important to note.
  • Although CAF can certainly affect the immune response, it is still unknown whether targeting both tumor-derived and CAF-derived signals would be more beneficial as an anti-tumor strategy and more work needs to be done to differentiate these two sources.
  • It is also important to mention as a discussion point whether CAF immunomodulatory functions are organ specific.
  • In terms of CAF heterogeneity, a recent study by Vickman et al (DOI: 10.1002/pros.23929) suggest that specific CAF subpopulations might interact with specific types of myeloid cells in prostate cancer. A small discussion about the need to better understand CAF heterogeneity in the context of immune evasion should be included.
  • It is important to note in the “Therapeutic Implications” section that many of the drugs that are being used to target TME are not necessarily blocking CAF-only signals. FAP perhaps is one of the few CAF-specific molecules. These molecules have been shown to be expressed/secreted by a variety of cell types including cancer cells and recruited inflammatory cells. In this regard it may actually be an advantage to block an specific target from multiple sources.
  • Table 3 and 4. Please modify the columns accordingly to accommodate the full word (or number) in each line.

Reviewer 3 Report

The review is timely - there is currently much interest in characterising the cancer-associated fibroblast population and understanding the regulation of different phenotypes for targeting as part of cancer therapy, particularly immunotherapy. CAF have been shown to modulate many aspects of the immune response and interact with many immune cell types; these are summarised briefly in the paper - but (perhaps understandably given space restraints) some sections lack detail, and perhaps it would have been better to ficus on CAF affects on T-cells rather than cover all immune cells, which tends to read as a rather dry list. 

A second aspect which needs more detail is the concept of CAF heterogeneity. A number of single cell RNA sequencing studies have now been published which have identified different CAF subtypes (eg iCAF) and some further consideration should be given to the functional role of these, as well as 'normal' phenotypes which may be tumour suppressive.

The section on clinical trials using CAF-targeting strategies is informative and useful, although some of the recent preclinical work looking at cross-over therapies from the fibrosis literature has been omitted  (eg. pirfenidone, nintedanib, NOX4 inhibitors)

Round 2

Reviewer 3 Report

The authors have addressed my previous comments satisfactorily - I have no additional requests